# Recent Advances in Lossy Mode Resonance-Based Fiber Optic Sensors: A Review

**DOI:** 10.3390/mi13111921

**Published:** 2022-11-07

**Authors:** Satyendra Kumar Mishra, Akhilesh Kumar Mishra, Paola Saccomandi, Rajneesh Kumar Verma

**Affiliations:** 1Department of Physics, Central University of Rajasthan, NH-8 Bandarsindri, Ajmer 305817, Rajasthan, India; 2Department of Mechanical Engineering, Politecnico di Milano, 20156 Milan, Italy; 3Centre of Optics, Photonics and Lasers, University of Laval, Quebec City, QC G1V 0A6, Canada; 4Department of Physics, Indian Institute of Technology Roorkee, Roorkee 247667, Uttarakhand, India; 5Department of Physics, University of Allahabad, Prayagraj 211002, Uttar Pradesh, India

**Keywords:** fiber optic sensors, lossy mode resonance, surface plasmon resonance, sensitivity, thin films and sensors

## Abstract

Fiber optic sensors (FOSs) based on the lossy mode resonance (LMR) technique have gained substantial attention from the scientific community. The LMR technique displays several important features over the conventional surface plasmon resonance (SPR) phenomenon, for planning extremely sensitive FOSs. Unlike SPR, which mainly utilizes the thin film of metals, a wide range of materials such as conducting metal oxides and polymers support LMR. The past several years have witnessed a remarkable development in the field of LMR-based fiber optic sensors; through this review, we have tried to summarize the overall development of LMR-based fiber optic sensors. This review article not only provides the fundamental understanding and detailed explanation of LMR generation but also sheds light on the setup/configuration required to excite the lossy modes. Several geometries explored in the literature so far have also been addressed. In addition, this review includes a survey of the different materials capable of supporting lossy modes and explores new possible LMR supporting materials and their potential applications in sensing.

## 1. Introduction

The exponential progress in optical fiber technology has retriggered the sensing capabilities of optical fiber-based devices [1,2,3]. These devices have exceptionally advantageous features such as a resistance to electromagnetic (EM) interference, a light weight, remote sensing capability, ease of design, the possibility of wavelength multiplexing, and a high sensitivity with wide detection range [4,5]. The collaboration of nanofilm coating and optical fibers turns out to be an important aspect in designing fiber optic sensors (FOSs) with widespread applications in physical, medical, and biochemical sensing [6,7,8]. Hence, by making use of the advantageous features of optical fibers and certain resonance techniques, especially lossy mode resonance (LMR) [9], the area of FOSs reaches a level where miniaturized low-cost devices can be made easily achievable for fast and consistent sensing applications. In this context, prism/optical fiber-based sensors plays a pivotal role in utilizing many of these beneficial properties. Prism-based optical sensors need a cumbersome experimental setup having larger spot of detection which binds their utility for sensing the nanosamples. On the other hand, optical sensors engaging an optical fiber core as a substrate offer several advantages such as high sensitivity, low weight, and easy handling, besides having high precision, flexibility, compactness, and easy transportation and being operative in hazardous environments. Due to these valuable features, FOSs are employed as a substitute for conventional optical sensors. The convention of these FOSs for the detection of chemical concentration has been described in the literature since the 1960s [10,11,12,13]. These FOSs have also found usage in environmental monitoring [14,15,16,17,18], biochemical sensing [19,20,21,22], and many other applications [23,24,25]. Since last decade, a lot of research has been carried out in the field of FOSs, utilizing different fabrication techniques and sensing principles [26,27,28,29,30]. Out of them, FOSs based on optical resonances (such as LMR) have drawn much attention owing to their better sensitivity and versatility. The study of the evolution of FOSs utilizing the LMR technique, leading to better performance parameters, is the source of motivation for the present review article. The coating of nanofilms having a complex refractive index (RI), i.e., a lossy nanocoating on the optical waveguides, results in the generation of several attenuation bands in the transmission spectrum, which is described as the mode coupling of the optical waveguide modes and the lossy modes of the thin semiconducting film [31]. These lossy modes are the guided modes having a complex effective index, also named as the long-range mode or leaky modes in the past [32,33]. The realization of LMRs utilizing thin absorbing films was investigated with the EM theory [34,35]. For the designing of a LMR-based FOS, usually a small sensing region, by detaching cladding layer from the optical fiber core, is considered. Subsequently, a thin film of semiconductor/conducting metal oxides (CMOs)/polymer is coated over the fiber core, for sensing the biochemical analytes. As soon as the light launched from one end of the optical fiber reaches the sensing region, in the form of the evanescent tail generated due to the phenomenon of total internal reflection, the sensing layer causes a variation in some property of the incident light because of the excitation of lossy modes and, hence, can witness a small change in the surrounding medium RI (SMRI). The equally important technique used so far is surface plasmon resonance; however, there are few similarities and contrasts between surface plasmon resonance (SPR) and LMR. This has regularly created doubts and compelled researchers to become perplexed about the two, until the first application of LMR was invented [34,36,37]. Villar et al., in 2010, utilized indium tin oxide (ITO) to study the concept of LMR both experimentally and theoretically [34]. Subsequently, a lot of fiber optic LMR sensors have been presented in recent times, by depositing various CMO films, such as ITO [38], indium oxide (In_2_O_3_) [39], titanium dioxide (TiO_2_) [40], aluminum-doped zinc oxide (AZO) [41], tin oxide (SnO_2_) [42], zinc oxide (ZnO) [43], etc., and polymers as well [44]. These materials exhibit more viabilities than the thin film of metals, due to their extraordinary structural, electronic, and surface properties, besides a larger band gap with tunable conductivity [45,46]. The capability of CMOs, to act as a coating/lossy material in visible and near-infrared (IR) regions, plays a crucial role in fiber optic sensing technology. The sensing applications of LMR includes pH sensors [47,48,49], antibody sensors [50,51,52,53], relative humidity sensors [54,55], liquid salinity sensors [56], gas sensors [57], and volatile organic compound sensors [58]. It is expected that, like SPR sensors, sensors based on the LMR technique will also gain popularity among the scientific/industrial community. 

The purpose of this review is to draw attention to the possibilities of using LMR-based FOSs for various physical, chemical, and gas-sensing applications, as well as to provide a comprehensive review of the promising results obtained in this field. This review includes works where the LMR phenomena have been employed for these sensing applications as well as other research papers that describe some fascinating sensing materials with a high potential to be used in the development of LMR-based FOSs.

In this review article, starting from the discussion of the basic principle and characteristics of LMR-based sensors, we have provided a detailed literature review of and the current progress in making LMR-based sensors. In Section 2, we discussed in detail the basic principle of lossy mode resonances, LMR-generation conditions, and their characteristics. Besides, we have also discussed the similarities and differences between LMR and SPR excitations in this section. Section 3 includes a review of the different structures of the optical fibers, i.e., different designs that have been considered to make LMR-based sensors so far. These structures include straight core, tapered (uniform and truncated), U-shaped, and side-polished D-shaped optical fibers. In Section 4 and Section 5, we explored the materials that have the potential to excite the lossy modes, for sensing applications. Finally, the conclusions and possible future scopes are provided in the last section. The authors believe that this review article will pave the way for people who want to begin their research work in the speedy, developing area of lossy mode resonances for sensing applications.

## 2. Basic Principle and Characteristics

### 2.1. Fundamentals

As the light propagation in a waveguide becomes affected, when a thin layer is deposited over it and if the RI of that coated material has a complex value, then certain attenuation bands are produced due to the lossy nature. Depending on the type of the material, waveguide, and sensing medium, the resonance techniques can be classified in the categories of SPR and LMR, as shown in Table 1. For SPR to occur, the real part of the dielectric constant of the coated film ($\varepsilon_{m}^{\prime }$) should be negative and higher in magnitude than its own imaginary part ($\varepsilon_{m}^{\prime \prime }$) as well as the permittivity of the sensing analyte ($\varepsilon_{s}$). On the other hand, LMR is achieved if the real part of the dielectric constant is positive and higher in magnitude than its own imaginary part. Moreover, if we use metallic nanoparticles (NPs) instead of a thin metal film, the interaction of the incident light with the metallic NPs (with a size that is comparable to the wavelength of the incident light) causes the coherent localized oscillations of free electrons, named as localized surface plasmons. Unlike SPR, in which coherent oscillations are confined within the metallic film, the coherent oscillations of electron clouds are localized in the metallic NPs, in the case of localized SPR (LSPR).

The phenomenon of LMR occurs due to the coupling of optical waveguide mode and a particular lossy mode of the coated film at a finite thickness. The mode coupling mainly depends on substantial overlapping between the mode fields and the state of phase matching [59]. The state of phase matching is attained when the propagation constant (only real part) of both the waveguide mode and lossy mode matches [60,61]. When lossy modes are near the mode-cutoff condition, both the conditions are satisfied, and lossy modes start to guide through the coated film. The cutoff condition is further characterized by two major parameters, viz. the wavelength of the incident light and the thickness of the coated film [34]. As a result of lossy mode excitation, sharp minima in the spectrum of the transmitted light are generated due to the maximum transfer of energy from the optical waveguide modes to the lossy modes, for a finite value of film thickness at several wavelengths, as depicted in Figure 1. This shows that multiple LMRs can be generated in the spectrum, if we increase the thickness of LMR-supported materials. In Figure 1c, the dip in the transmission spectrum, which is at the higher side of the wavelength region, is due to the first lossy mode excitation, called the first LMR curve. Similarly, dips on the lower side of wavelength are due to the excitation of the second and third lossy modes and, hence, are called the second and third LMR curves, respectively. A detailed analysis of these LMRs reveals that at higher wavelengths, no mode is guided inside the coated material, but if we tune to lower wavelength values, the first lossy mode is close to the cutoff condition, and, hence, the first LMR is visible in the transmission spectrum. If we further tune to lower wavelength values, a second lossy mode overcomes the cutoff condition and, hence, becomes guided in the coating. The observed LMR is called a second LMR, because this is due to the guidance of the second lossy mode in the thin-film coating, and a similar explanation is valid for the subsequent LMRs in the transmission spectrum. It should be noted that once a mode becomes guided in the lossy film, it recovers its original state. Moreover, these LMRs are strongly dependent on the coated film’s thickness, and, hence, the number of LMRs in the transmission spectrum can be increased just by increasing their thicknesses, but their sensitivity will always be lower than that of the first LMR. Similar to SPR, LMR-based sensors can function with two kinds of schemes, viz. wavelength interrogation and angular interrogation. 

### 2.2. Resemblances and Differences between SPR and LMR

It becomes necessary to identify the similarities and differences between the two kinds of resonances, i.e., SPR and LMR. Like SPR, LMR also depends on various parameters such as the SMRI, thickness, and dielectric permittivity of the coated film. Further, both techniques use the coating of thin films as a sensing layer to excite surface plasmon (SP) modes/lossy modes, and the substances or medium to be detected are placed around that coated film. Both these resonance techniques can work on the single sensing principle and utilize the Kretschmann–Raether configuration. Moreover, in both techniques, it is possible to use additional coating layers adhering to the resonance supporting layer for the detection of various sensing analytes [43]. Finally, the attenuation bands in the spectra are similar in both kinds of resonances, showing a sharp dip in intensity at the resonance parameter, and these resonance dips are highly sensitive towards any change in the SMRI. 

In terms of differences, the LMR technique supports a wide range of materials including the thin film of CMOs or polymers for the fabrication of a fiber optic sensing probe, whereas SPR needs the thin film of metals for the excitation of SPs. The CMOs are relatively economical, and researchers have easy access to them, which is also beneficial for the large-scale production of LMR-based FOSs. To achieve the SPR phenomenon, only transverse magnetic (TM)-polarized light is used, whereas in LMR both TM and transverse electric (TE)-polarized light can be considered, which substantially simplifies the experimental setup. In case of SPR technique, while launching light, the typical range of the incidence angle is 40° to 75°; however, LMRs are generated by launching the light at the grazing angle incidence, i.e., the angle approaching to 90° [62,63]. Thus, most experimental work on LMRs utilizes an optical fiber core instead of an optical prism because, in the latter case, it becomes very difficult to direct light at around 90° [64]. 

Further, a notable difference is that unlike SPR, which has only one resonance dip, the LMR spectrum produces multiple resonance dips and can be tuned by varying the thickness of the coated film too, without any modifications in the optical fiber geometry. This means that when a solo parameter is measured, the accuracy of the measurement can be guaranteed by obtaining information from multiple resonance dips, and, when multiple parameters are measured, one can obtain a different parameter’s information from the different resonance dips, which contribute to signal demodulation. This property makes the LMR technique more suitable for fabricating multi-peak sensors, with a higher sensitivity and better accuracy [64]. Apart from this, the resonance curves are broader and shallower in LMR-based FOSs than those in SPR-based FOSs, leading to a decrease in the resolution of the measurements and, hence, a decrease in the full width at half maximum (FWHM) and figure of merit (FOM) values. This may be because both TE- and TM-polarized light can generate LMR, i.e., LMR_TE_ and LMR_TM_. However, due to the smaller value of the imaginary part of the thin-film RI, the LMR_TE_ and LMR_TM_ are located at different wavelengths, and the largest gap can reach several hundred nanometers.

In addition, the resonance dips in the SPR technique are located in the visible region of the optical spectrum in general; however, the LMR dips lie in the visible or near-infrared (IR) regions. Currently, the information transmission of optical fibers lies in the near-IR region, favoring LMR-based FOSs.

## 3. Geometrical Considerations

To enhance the performance of fiber optic LMR sensors, various geometries and configurations of optical fibers have been realized recently, such as straight core optical fibers [39,42,43,55,65,66,67,68], tapered fibers [69,70,71,72,73,74,75], U-shaped fibers [76], D-shaped fibers [77,78,79,80,81], etc. In this section, we provided a detailed explanation of various fiber geometries and the steps involved during their fabrication processes. 

### 3.1. Straight Core Optical Fibers 

It is now well-known that the light propagating inside the optical fiber is mainly comprised of two components, i.e., the guided field in the fiber core and the evanescent field in the fiber cladding. In an optical fiber, the magnitude of the evanescent field decays to almost zero within the cladding, and, hence, the light propagating inside the fiber does not interact with the medium surrounding the optical fiber. This is because optical fibers were initially used for light propagation and to utilize them in sensing, it is required that the light should interact with the surrounding medium. The easiest and most reliant way to fabricate a fiber probe is to use straight- or uniform-core plastic-cladded-silica optical fibers, such that the removal of the plastic cladding may lead the evanescent field to interact with the sensing or surrounding medium and, also, raise the penetration depth [82]. For this, initially a small portion of cladding from the middle portion of the optical fiber is removed gently, and then the cladding-etched fiber core is washed continuously with acetone and distilled water to remove any ionic impurities. The schematic view of an unclad optical fiber is shown in Figure 2a. After this, using the different thin-film-deposition techniques such as the thermal evaporation technique, the e-beam technique, dip coating, the layer by layer (LbL) method, sputtering, etc., a thin film of CMO is coated uniformly over it, for its usage in LMR sensing [83]. A detailed transmission setup of a fiber optic LMR sensor is depicted in Figure 3. Sanchez et al. fabricated an optical fiber refractometer utilizing a thin film of SnO_2_ on a uniform fiber optic probe [42]. They employed the sputtering technique to coat a continuous and uniform thin film of SnO_2_ (with thicknesses of 55 nm, 594 nm, and 700 nm) on a multimode fiber having a core diameter 200 µm, and the same was confirmed via scanning electron microscopy (SEM) analysis of the fabricated sensing probe. The experimental absorption spectra were obtained by using halogen light as the excitation source launched from one face of the fiber, whereas another end of the fiber was attached to a spectrometer. The experiment was performed for a wavelength region of 400–1700 nm and to measure the wavelength shift of LMR absorption peaks, different solutions of glycerol–water were prepared such that their RI values fell between 1.33–1.473.

Later, Usha et al. theoretically designed a fiber optic LMR sensor on the concept of geometrical optics [84]. The study was carried out for different thicknesses of a ZnO layer coated over the unclad portion of a uniform fiber core, and the thicknesses were optimized such that multiple LMRs can be achieved in the transmission spectrum. Theoretical analysis concluded that a fiber optic LMR sensor employing a thin film of ZnO shows better sensitivity values and, hence, can also be investigated to observe the compatibility of a nano LMR sensor. More recently, an experimental investigation of LMR generation was done by depositing a thin film of ZnO over a 4 cm unclad multimoded optical fiber [67]. For this, the sputtering technique was employed to coat an AZO film of different thicknesses. The experimentally observed absorption spectra revealed that multiple absorption peaks of LMR were achieved with an increase in thin the film thickness, and the corresponding second, third, and fourth LMRs were attained at wavelengths of 1090 nm, 670 nm, and 510 nm, respectively. The first LMR comes from beyond the working wavelength range and, hence, cannot be measured. Therefore, the absorption spectra, corresponding to the second, third, and fourth LMRs, were measured as a function of the wavelengths for the different SMRIs. It was found that each LMR shifted to a higher wavelength value with an increment in the SMRI, and the maximum wavelength shift is for the second LMR when compared with the rest of the LMRs. Thus, the fabrication of fiber optic LMR sensors utilizing these deposition techniques may lead to several new applications because of the lower cost and the intrinsic properties of such sensors. In 2019, Wang et al. proposed a fiber optic LMR sensor fabricated with a thin film and SnO_2_ NPs, using the LbL deposition technique [68]. The optical fiber, having a core diameter of 600 µm, 0.37 as its numerical aperture, and 1.5 cm as the length of its unclad region, was considered for depositing a thin film over it. The used SnO_2_ film possesses a thickness of 40 nm, and NPs having a particle size from 50–70 nm further enhanced the sensitivity of the LMR sensor when compared with the case of one without NPs. The designed LMR sensor shows good repeatability of the sensing probes, and the theoretical/experimental sensitivity values were attained as 5334 nm/RIU and 4704 nm/RIU, respectively. Furthermore, the LbL method used was successfully implemented for the fabrication of SnO_2_ thin films in the design of fiber optic LMR sensors, which also predicts the crucial sensing applications of SnO_2_ in LMR.

### 3.2. Tapered Core Optical Fibers

Tapered optical fiber is a special type of geometry that strongly exposes the exponentially decaying evanescent field, to interact with the sensing analytes. Tapering the fiber also increases the magnitude of the evanescent field and penetration depth, which leads to the potential applications of optical fiber in the area of sensing [85,86,87]. The tapering of optical fibers can be done via etching the cladding portion from the middle of the fiber and then tapering the unclad fiber core [88]. The fabrication process for tapering the fiber involves several methods and theories, such as electric arc discharge or heating the fiber core with a moving burning flame. There are several geometric techniques to enhance the evanescent field amplitude of FOSs; however, tapering the fiber core is one of the most useful techniques to be used in LMR-based sensing applications in various laboratories and industries. The typical view of the commonly used LMR-based tapered fiber probes is shown in Figure 2b,c. In Figure 2b, the transition tapered region acts as a sensing region, whereas in Figure 2c the uniform fiber core placed between two identical tapered regions act as the sensing region. Thus, in a continuous tapered fiber, the diameter of the fiber core decreases gradually, reaches a constant diameter waist region, and then returns to its original position.

In 2012, Socorro et al. designed a fiber optic LMR sensor utilizing a thin polymeric film coated over a tapered single-mode fiber, with an initial core diameter of 8.2 µm [70]. The tapering procedure was done with the help of a traveling burner that contains a tiny flame, through which the pulling process can be controlled. After tapering, the fiber parameters were considered to be around 2 µm for the core diameter, 10 mm for the length of the uniform waist, and 3 mm for the length of the transition taper region. Later, a thin polymeric film was coated over the fiber cladding, by employing the LbL electrostatic self-assembly method. In order to observe the LMR transmission spectra, five different PBS solutions were prepared with pH values ranging between 4.0 and 6.0. From the obtained results, it was found that the resonance wavelength (RW) shifted to a higher value with the decrease in pH values, and also the LMR dips lie in the near-IR region. This work presented the various wavelength values at which the sensor stops after dipping in each pH concentration. The work made an important conclusion: sensitivity value can be enhanced by reducing the waist diameter of the tapered fiber core, and, hence, the designed LMR sensor can be useful for studying the behavior of LMR-based pH sensors. Vikas et al. presented the design considerations of LMR-based tapered FOSs utilizing ITO and AZO thin films [74]. They considered a step index multimode fiber, with a numerical aperture of 0.22 and a core diameter of 600 μm. Then, a small portion of the cladding (0.5 cm) was removed from the middle of the optical fiber, and the unclad portion of the fiber was assumed to be tapered. Figure 4a depicts the structural view of a used tapered-core fiber optic probe, which only shows the tapered region; however, a complete probe will follow the mirror image of this. Figure 4b shows the variation of the taper radius *s*(*z*), with distance *z* for five taper profiles, and the graphical presentation of the sensitivity with the TR along with the five taper profiles is shown in Figure 4c,d for the first and second LMRs of ITO, respectively. From the obtained results, it was found that with a rise in TR, the sensitivity enhances for both LMRs corresponding to the taper profiles. Moreover, it was noticed that out of five different taper profiles, the maximum sensitivity values were achieved for the exponential–linear taper profile, whereas the quadratic profile possessed the least values due to the minimum waist diameter of the exponential–linear taper profile. The above results concluded that the highest sensitivity value for the exponential–linear taper profile was observed as 12,005 nm/RIU for the first LMR at a taper ratio (TR) of 1.7, and also the first LMR showed higher sensitivity values than the second LMR. In another work, a fiber optic LMR sensor having a uniform core positioned between two similar tapered fiber regions was studied theoretically [75]. Here, the middle uniform core coated with ITO or AZO acts as a sensing region; however, the tapered fiber regions were considered in order to bring the incident angle close to the critical angle, as depicted in Figure 5. In this, a thin film of metal oxide becomes coated over the uniform fiber core, and, subsequently, this layer was kept in direct contact with the sensing region. For ITO, the maximum sensitivity values for the first and second LMRs were 18,425 nm/RIU and 825 nm/RIU, respectively, at a TR of 1.6, and the corresponding values for AZO were 790 nm/RIU and 350 nm/RIU, at a TR of 2.0. A further increment in the TR results in a distorted LMR spectra, and the sensitivity cannot be calculated. This enhancement in the sensitivity can be clarified from Figure 5: using taper region 1, the angle of the bounded rays into the optical fiber becomes closer to the critical angle. This can be obtained by taking the least output diameter of the taper fiber core that causes the entire bounded ray to propagate inside the sensing region. Then, after crossing the sensing region, the ray will enter into tapered region 2, which further converts the angle of the bounded rays to their original values, in order to get the complete bounded ray at the output end of the optical fiber. If we choose the minimum value of the core diameter, then no ray will escape out, and a majority of the rays will propagate with the angle closer to the critical angle, and, hence, the evanescent wave (EW) will penetrate with a higher penetration depth.

### 3.3. U-Shaped Optical Fibers

This is a special kind of geometry that is formed by the bending of the optical fibers, which leads to an increase in the evanescent field’s amplitude, and, hence, an increase in the penetration depth [89]. It is verified in the literature that a U-shaped fiber probe possesses better absorption and sensing capabilities when compared with a uniform-core fiber optic probe of identical dimensions [90]. This is because in a uniform-core fiber optic probe, the angle of ray propagation remains constant, and, thus, the number of reflections and absorption coefficient depends only upon the core radius; whereas, in the case of a U-shaped probe, the angle of ray propagation is not fixed and instead decreases. Only a few articles related to LMR sensors designed on a U-shaped probe are available. Paliwal et al. reported the design considerations of a U-shaped LMR sensor both theoretically and experimentally [76]. To fabricate a U-shaped probe, initially a small cladding portion was detached from the middle portion of the optical fiber, and then the cladding-removed portion was bent in the form of the English letter U. After, this a thin film of a ZnO layer is coated over a U-shaped optical fiber, with various bend diameters, to generate the LMR curves. To further simplify the model, the study neglected the effect of the coating on the inner bend because it does not contribute much to the coupling of the evanescent wave with the lossy modes. The wavelength interrogation scheme was employed to measure the transmitted light at the output end of the U-shaped optical fiber. To utilize the sensing probe as a LMR sensor, the experimentally observed sensitivity values were calculated in terms of the LMR shift, absorbance, and FWHM, which vary in accordance with the SMRI. If we see the effect of the bending radius on the sensitivity, it was revealed that the sensitivity increases with a decrease in the bending radius, but after a specific value the sensitivity values start decreasing. This is because of the overlapping of the evanescent field, which results in a power transfer between the two arms of the U-shaped probe, without propagating through the bent region. It was observed that the U-shaped fiber probe modified with the deposition of a ZnO film results in a six-fold increment in sensitivity, when comparing with the results obtained for straight-core fiber optic LMR sensors. The designed U-shaped geometries for LMR sensors can find important applications in biomedical sensing [91]. 

### 3.4. D-Shaped Optical Fibers 

A D-shaped optical fiber coated with a nanofilm has shown better sensing capabilities in various domain of sensors. It is achieved by side polishing the optical fiber segment and then fabricating it with the deposition of a thin film over it, which results in the better interaction of the propagating light and the coating thin film [36]. Related to those of a LMR sensor, the sensitivity values of D-shaped optical fibers coated with a nanofilm were achieved in the range of 3000 to 8000 nm/RIU [37,77], but, in recent years, the highest sensitivities of 15,000 nm/RIU and 300,000 nm/RIU have been achieved when surrounded with water and a silica RI, respectively [92]. Thus, fiber optic LMR sensors designed on a D-shaped optical fiber turn out to be a promising structure in RI-sensing applications. In a continuation of this, highly sensitive LMR sensors were fabricated experimentally with the deposition of ITO film over a D-shaped optical fiber for both TM- and TE-polarized light [77]. 

Figure 6a represents the experimental setup used for the fabrication process of ITO and a schematic view of the D-shaped fiber. The used optical fiber is a single-mode D-shaped fiber polished from one side, having a core diameter of 8 µm, and a side-polished length of 1.7 cm. Later on, a particular fiber was coated with ITO utilizing the DC sputtering technique and then placed in direct contact with a sensing solution, having different RI values prepared via the addition of glycerol in water. This experimental setup measures the response of the output power at different wavelength values (1300 nm and 1500 nm), in terms of their deposition time, with an interval of 1 s, as shown in Figure 6b. From the obtained spectra, we noticed that two different wavelength values enable us to measure the shift in the maximum absorption peaks for various coating thicknesses. It was also observed that for higher-order LMRs, the separation between LMR_TE_ and LMR_TM_ is decreased, and, at a particular value of film thickness, LMR_TM_ approaches to LMR_TE_. The maximum sensitivity corresponding to LMR_TM_ was 8742 nm/RIU, with the SMRI ranging from 1.365 to 1.38, which is much higher than in the case of a SPR-based FOS. In 2017, Villar et al. reported that the FWHM of the LMR curve for a sensor based on a D-shaped optical fiber can be minimized by optimizing the width, thickness, and refractive index (imaginary part) of the coating [78]. Moreover, it was concluded that the need for using a polarizer can be avoided by fabricating a device in which the resonance bands for both TE- and TM-polarized light are located at identical wavelengths. Zubiate et al. proposed a LMR device for the detection of a C-reactive protein (CRP) realized on a D-shaped optical fiber [53]. The sensor uses the ITO as the LMR’s active material to be coated over the planar region of the side-polished D-shaped fiber, in order to attain the high sensitivity and selectivity of a LMR sensor. The sensing layer designed with ITO in that work was further used to measure the particular interactions among CRP–aptamer and CRP. The obtained results showed that the RW shift for each CRP concentration leads to the better stability and monotonous response of the sensor. Moreover, a stable response of the device was noticed during immersion in CRP and a buffer solution. The study confirmed that the shift in the resonance wavelength was proportional to the change in the SMRI because of the better interaction of the adapter chain and protein. The variation of the LMR shift with various concentrations of CRP, urea, and creatinine showed that a shift in RW is insignificant when the device was dipped in a solution of urea and creatinine, which means the designed sensor was highly selective towards the detection of CRP only. More recently, Wang et al. numerically anticipated a fiber optic LMR sensor employing a D-shaped optical fiber with the bilayer deposition of TiO_2_ and HfO_2_ [79]. The used optical fiber had a core diameter of 15 µm, a hole diameter of 50 µm, and an air holes’ thickness of 10 µm, and the distance of its adjacent air holes was taken as 2 µm. The theoretical analysis for the study of EM field distribution was done using the finite element method in COMSOL Multiphysics software. The study reported that a minute change in the SMRI results in a modification in the phase-matching condition and, hence, produced different lossy spectra. Thus, a little variation in the SMRI can be measured accurately by precisely locating the shift in peak wavelength. For the designed D-shaped LMR sensor, the maximum sensitivity was achieved up to 140,000 nm/RIU, with the SMRI = 1.395. In 2020, Tien et al. experimentally demonstrated a novel RI and salinity sensor based on the LMR technique, which comprised the combined effect of the side-polished D-shaped fiber and radio frequency sputtering techniques [81]. The sensor utilized the thin film of a gallium-doped zinc oxide (GZO) having a thickness of 67 nm, which was then coated over a side-polished D-shaped optical fiber. From the obtained results, it was summarized that a coating of GZO on D-shaped optical fibers is capable of generating LMR absorption peaks, shifting to a higher wavelength value with an increase in the SMRI. The maximum value of sensitivity was achieved up to 3637 nm/RIU, in the RI ranging from 1.333 to 1.392. We studied the different geometries of LMR-based FOSs, and it was concluded that the designed fiber optic LMR sensors with a nanofilm coating possess higher sensitivity values and are capable of sensing several chemical species. 

## 4. Materials Supporting Lossy Modes for Sensing Applications

Unlike SPR, the LMR phenomenon is supported by a wide range of LMR active materials including various metal oxides such as indium tin oxide (ITO), indium oxide (In_2_O_3_), zinc oxide (ZnO), tin oxide (SnO_2_), titanium oxide (TiO_2_), etc., and polymers as well. In recent years, the gas-sensing mechanism of metal oxides has been extensively investigated, and it is believed that the initial process is the diffusion of the analyte gas from the surrounding medium to the semiconducting metal oxide surface [93]. The second process involves a charge-transferring interaction between the analyte gas and the metal oxide surface. This depends on the gas adsorption, the change in charge-carrier concentration in propinquity to the oxide surface, and the surface reactions [94]. Lastly, the working temperature of the sensor plays a crucial role in the formation of reactive species and the chemisorbed reactive oxygen species’ role, either in the formation of reactive species or in the chemisorbed reactive oxygen species/ions, per the following reactions [95,96]:$$O_{2\mathrm{ads}}+e^{-}⇌O_{2\mathrm{ads}}^{-}\;(<100\;{}^{\circ}C)$$
$$O_{2\mathrm{ads}}^{-}+e^{-}⇌2O_{\mathrm{ads}}^{-}\;(100\;-300\;{}^{\circ}C)$$

The formation of these oxygen ions results in the capture of electrons from the conduction band of the surface layer, determining a change in the conductivity of the metal oxides [96]. In addition to these metal oxides, there are certain polymers, i.e., polyacrylic acid (PAA), polyallylamine hydrochloride (PAH), and polystyrene sulfonate (PAS), which have also been employed in LMR-based sensing in recent times. Therefore, the present section discusses the detailed explanation of each LMR-supported coated material and their further utilization in various sensing schemes, as found in the literature.

### 4.1. Indium Tin Oxide (ITO)

ITO is one of the most recognized CMOs, which has a wide range of applications in optics and electronics [97]. It is mainly a compound of indium oxide and tin oxide, having different composition ratios that affect their electrical and mechanical properties. ITO films are chemically stable in nature and, hence, can simply be coated over an optical fiber core via several deposition methods. The dispersion curve depicted in [44] confirms that ITO satisfies the generation condition of both SPR (at a higher wavelength, from 0.4–1.3 µm) and LMR (at a wavelength beyond 1.3 µm), as discussed above. Moreover, its conductivity increases further by increasing the thickness of ITO and, hence, results in a better sensitivity for the LMR sensor, which makes ITO a promising candidate in various sensing applications. Zamarreno et al. experimentally designed a FOS utilizing a thin film of ITO coated over a uniform fiber core in the IR region of the optical spectrum [38]. The ITO layer allowed the coupling of light from the optical waveguide layer to the ITO/sensing medium as a function of the SMRI. The study showed that a maximum sensitivity of 3125 nm/RIU is achievable, with the utilization of an ITO film that quickly senses the analyte. In another work, a LMR-based FOS was designed for the detection of thrombin, by utilizing an aptamer receptor [98]. The designed LMR sensor consisted of a novel aptamer–polymer structure fabricated over an ITO-coated optical fiber core employing the LbL electrostatic self-assembly method. The variation in the RI of thrombin, which acts as a sensing region, results in the wavelength shift of the LMR curves, and, hence, the designed configuration detected even low (100 nM) and very high (1 µM) concentrations of thrombin. The fabricated LMR device showed a better response time with a maximum shift of 3 nm in the RW. Another application of ITO-coated LMR-based FOSs was reported for the detection of relative humidity (RH) [99]. In that study, a thin coating of ITO was studied over a cladding-etched multimoded optical fiber, with a core diameter of 200 µm to sense the relative humidity. The sensor, designed with ITO, attained a sensitivity of 0.283 nm/%RH for RH detection, and this value was further enhanced to 0.833 nm/%RH by coating an addition layer of PAH–PAA over the ITO. In 2013, Corres et al. proposed a LMR-based tunable filter with the multilayer deposition of ITO/PVDF/ITO over a plastic-clad silica (PCS) multimoded optical fiber [100]. The initially used ITO layer generated the LMRs in the spectrum and also acted as a first electrode; the PVDF layer tuned the filter, and the second ITO layer allowed for the measurement of the variation in its RI as a function of the applied voltage, in order to attain a high sensitivity, i.e., 0.4 nm/V. In a similar study of electrochemical processes, an ITO layer was coated over the fiber core and was applied as a working electrode in a cyclic voltammetry setup [101]. The results concluded that any change in applied voltage corresponding to the ITO surface resulted in a substantial variation in the LMR response. Additionally, a fiber optic LMR sensor coated with ITO film can be considered as a dual-domain sensor, when operated with the optical and electrochemical processes [102]. Soon after this, an ITO-coated RI sensor based on LMR was designed, employing angular and wavelength interrogation techniques [103]. The authors reported the maximum value of sensitivity as 4652 nm/RIU, by inserting an additional low-index matching layer. In a similar study, it was shown that a particular thickness of ITO can generate both SPR and LMR in the wavelength range of 450 to 2000 nm, which can be further used in optical sensing [104,105,106,107,108,109]. Recently, a fiber optic LMR sensor coated with ITO thin film was used for the detection of hydrogen (H_2_) gas sensing [57]. The reported work compared the performance of a designed sensor by fabricating three different fiber optic probes, i.e., a fiber core coated with ITO film, ITO nanoparticles, and a combined layer of ITO film/NPs. A schematic view of an experimental gas-sensing system is depicted in Figure 7. It was observed that the corresponding values of sensitivity were 0.32, 0.58, and 0.71 nm/ppm, respectively, showing that the additional layer of NPs over the thin film greatly enhances the sensitivity of the H_2_ gas sensor. Moreover, an ITO-coated fiber optic LMR sensor was utilized for the fabrication of a lubricant oil degradation sensor at room temperature [110]. The designed sensor was capable of sensing the oil degradation, by locating the wavelength shift in the LMR spectra as a function of the SMRI, and achieved a maximum sensitivity of 0.15 × 10^−3^ nm/h. 

In the same year, Torres et al. presented a Kretschmann-configuration-based LMR/SPR sensor coated with an ITO thin film over a fiber core [62]. It was reported that the ITO was capable of generating LMR resonances for both TE- and TM-polarized light, with sensitivity values of 700 and 1200 nm/RIU, respectively. On the other hand, the corresponding value of sensitivity for SPR was 8300 nm/RIU, for a change of 1.47 in the SMRI. Recently, fiber optic LMR sensors for real-time monitoring of ketoprofen were anticipated by fabricating a thin ITO film over 2.5 cm long photonic crystal fiber (PCF) [111]. It was revealed that the spectral response of an ITO-based LMR sensor is strongly dependent on the electrochemical modification of its surface by ketoprofen, which is effective for the real-time monitoring of ketoprofen. The maximum sensitivity values were attained as 1400 nm/M and 16,400 a.u./M for the RW and transmission shifts, respectively. Moreover, the working range of the designed sensor was 0.25–250 µgmL^−1^ for ketoprofen concentration, which can also be used for its detection at therapeutic and toxic levels. Another important application of an ITO-based fiber optic LMR sensor is the detection of femtomolar concentration in human serum, which was reported by Chiavioli et al., in 2018 [112]. The experimental study showed that the detection limit of a designed LMR sensor was enhanced by three times, when compared with the existing optical fiber configurations. More recently, a LMR-based FOS was theoretically realized by depositing a 90 nm ITO film over a tapered-core fiber optic probe using an *N*-layer matrix model with a wavelength interrogation scheme [74]. The thickness of the ITO film was optimized in order to attain a high RI sensitivity, and the results were compared with a SPR-based sensor with a similar fiber configuration. Since the first LMR is always more sensitive than the rest of the lower-order LMRs, thus, the calculation was completed only for the first two LMRs obtained in the transmission spectrum. Figure 8a depicts the transmission curves for a LMR-based FOS coated with an ITO film at a taper ratio of 1.0, for two different SMRIs. It is clearly visible that the corresponding LMR shift was higher in the case of the first LMR, with a variation of 0.002 in the SMRI, and the sensitivity values corresponding to each LMR were calculated separately, as shown in Figure 8b. Finally, the study concluded that the designed LMR sensor showed up to a 2.5-times increment in sensitivity, when compared with the SPR-based fiber optic configuration, and, hence, ITO comes out to be a better LMR-supported coating material, with its various sensing applications. Table 2 shows the detailed summary of an ITO-coated fiber optic LMR sensor, as discussed above.

### 4.2. Indium Oxide (In_2_O_3_) 

In_2_O_3_ is another CMO material that often displays transparency in the visible wavelength regime, which is mostly utilized in batteries, solar cells, and optical coatings [134]. In the early part of the 20th century, In_2_O_3_ was reported as being used for the detection of NO_2_ gas, with the chemisorption process over its surface having a concentration of up to 5 ppm [135].

The real and imaginary parts of the RI of In_2_O_3_ are depicted in Figure 9, which shows that In_2_O_3_ satisfied the condition of LMR in the plotted wavelength region [136]. The first application of In_2_O_3_ in LMR-based optical fiber refractometers was reported by Zamarreno et al., in 2010 [113]. The designed sensor showed a maximum RI sensitivity of 4068 nm/RIU, for a variation between 1.333–1.392 in the SMRI. In their another work, the authors presented the design considerations of a fiber optic LMR sensor utilizing an In_2_O_3_ film for both TM- and TE-polarized light, and the corresponding sensitivity values were 4255 nm/RIU and 4926 nm/RIU, respectively [39]. 

The study also considered the effect of coating thickness on the sensitivity of the sensor, and, for the same purpose, three different probes were fabricated of varied thicknesses, i.e., 25 nm, 49 nm, and 86 nm, over an optical fiber core. In a study where an ITO film was coated over an optical fiber core for the fabrication of a RH sensor, the employment of In_2_O_3_ for a similar purpose was also discussed [99]. A thin film of In_2_O_3_ with a thickness of 85 nm was deposited over a multimoded optical fiber, which resulted in two sharp LMR dips in the transmission spectra. This layer was then coated with a polymeric coating to enhance the response of the RH sensor. This resulted in a maximum sensitivity of 0.935 nm/%RH, corresponding to the TE-polarized mode, which was higher than in the case of an ITO-coated RH sensor (0.833 nm/%RH). Sanchez et al. demonstrated a highly sensitive LMR-based optical fiber refractometer, by depositing an In_2_O_3_ film over a cladding-removed fiber core via the DC sputtering technique [114]. Figure 10 depicts the absorption peaks observed from the device having different coating thickness (70 nm, 300 nm, and 690 nm) in the visible and near-IR regions of the wavelength. For the designed sensor, the maximum sensitivity was 5680 nm/RIU, corresponding to first LMR when the SMRI changes from 1.332 to 1.407. Further analysis of the In_2_O_3_ coating in LMR sensing was carried out theoretically, using modal analysis and experimental results that showed a better agreement with the theoretical results [35]. More recently, Verma et al. experimentally designed a LMR-based RI sensor with the deposition of In_2_O_3_ over a fiber core and compared the results with those obtained for SPR sensors [115]. Results confirmed that fiber optic probes having a 100 nm In_2_O_3_ deposition possess two times better the sensitivity than fiber probes having a silver (Ag) film. 

Finally, we concluded that the fiber optic LMR sensors based on an In_2_O_3_ coating found less utilization, when compared with ITO and other materials. This may be due to the presence of the more significant compounds formed by the additional doping of tin, gallium, and zinc, which possess better catalytic and electrical properties [137]. A detailed summary of the In_2_O_3_-coated fiber optic LMR sensor and its important applications are presented in Table 2.

### 4.3. Zinc Oxide (ZnO)

The next material that fulfils the generation condition of LMR is zinc oxide, or the compounds formed via doping in it. 

Zinc oxide belongs to the CMOs group, which has a better absorption band in the ultraviolet–visible region of the optical spectrum and, hence, turns out to be a promising material in optical applications [138]. Apart from this, it possesses various significant properties such as a high transparency, wide band gap, better thermal conductivity, and higher carrier mobility, which makes it a suitable material in the field of electronics and sensing [139,140]. The dispersion curves for ZnO are depicted in Figure 11, which confirm that it acts as a LMR active material in the visible region and satisfies the SPR condition in the near-IR region of the wavelength. The first LMR sensor that utilized ZnO as a transducing material was reported by Andreev et al., who observed a sensitivity value of 1700 nm/RIU for a variation between 1.333–1.450 in the SMRI [37]. However, the primary experimental investigation of a fiber optic LMR sensor utilizing ZnO as a thin LMR coating material was reported in 2015 [43]. The authors presented the fabrication and characterization techniques of fiber optic LMR sensors for hydrogen sulphide (H_2_S) gas detection. Fiber optic probes were fabricated by coating a thin film of ZnO and ZnO nanoparticles over an unclad portion of the fiber core. The ZnO layer used reacts with H_2_S, which resulted in the change of the dielectric constant of ZnO. To fabricate the fiber probe, a PCS multimode fiber with a core diameter 600 µm was used, and then a 12 nm ZnO film was coated over a cladding-etched uniform fiber core employing a thermal-evaporation-based vacuum coating unit. The fiber probe was then coated with ZnO nanoparticles by dipping it into solution of ZnO NPs. To realize the fabricated fiber probe as a LMR sensor, the experiments were performed with several concentrations (10–100 ppm) of H_2_S gas, and the corresponding LMR absorption spectra were recorded. The results showed that with an increase in H_2_S gas concentration, the absorbance and, hence, wavelength corresponding to the maximum absorbance increases. This is due to the modification in both the real and imaginary parts of permittivity, as well as the RI of ZnO when it comes in contact with H_2_S gas. The corresponding variation of the peak LMR wavelength with the H_2_S concentration revealed that the LMR wavelength shows a non-linear increment with an increase in the H_2_S concentration. Thus, it was concluded that the designed LMR sensor for H_2_S gas detection shows a better selectivity, and the same is verified by completing the experiment with different gases.

In another work, the combination of a ZnO thin film and nanorods was used for the fabrication of LMR-based H_2_S gas detection [116]. It was found that a fiber optic probe modified with an additional layer of ZnO nanorods over a thin film turns out to be more sensitive (4.14 nm/ppm) than a fiber probe without the ZnO nanorods (2.35 nm/ppm), for the detection of H_2_S gas. Apart from gas sensing, a novel fiber optic cortisol sensor was designed, by utilizing the combined effect of LMR and the molecular imprinting (MIP) of the ZnO nanocomposite layer and polypyrrole (PPY) [51]. Different solutions of cortisol in artificial saliva were prepared, having a concentration range between 0 and 10^−6^ g/mL. From the obtained results, it was concluded that a nanocomposite having a 20% concentration of ZnO in PPY gives the maximum value of sensitivity for the detection of cortisol. A similar kind of study was carried out for the diagnosis of urinary p-cresol, using nanocomposite of ZnO/MoS_2_ and a MIP polymer fabricated over a LMR-based optical fiber core [83]. The fabricated fiber optic probes were tested for the detection of p-cresol, with a solution that was prepared in artificial urine having a concentration range of 0–1000 μM. By optimizing the various fabrication parameters, the maximum sensitivity was 11.86 nm/μM, with a detection limit of 28 nM and a response time of 15 s. ZnO thin films are also utilized for the fabrication of fiber optic LMR pressure sensors [117]. In this work, the authors theoretically proposed the fabrication of a HfO_2_ overlay on top of a ZnO thin film, to attain the maximum sensitivity values. For the designed configuration, a sensitivity of 2 μm/MPa was detected when a LMR-based refractometer was inserted in a rubber block, with an RI that was dependent on the applied pressure. It is to be noted that some coating materials synthesized via doping in ZnO have also been studied in recent years, for the development of fiber optic LMR sensors. As studied in the case of ITO [74,141,142,143,144], a similar calculation was done for AZO, when coated over a tapered fiber core, for the realization of LMR-based RI sensors. For the proposed LMR sensor, the highest sensitivity values corresponding to the first and second LMRs were 515 nm/RIU and 235 nm/RIU, respectively. In an experimental study, AZO films were coated over a multimoded optical fiber to investigate the LMR effect [118]. The AZO films were coated by using a RF magneton sputtering technique, and the corresponding optical transmission spectrum with a variation in the SMRI was measured at the output end of the optical fiber. Figure 12 shows the variation of the RW, with a different concentration of isopropyl alcohol/glycerine, for the proposed fiber structure. For the designed sensor, a maximum value of RI sensitivity was observed to be 1214.7 nm/RIU, with a thin AZO film doped with 8% aluminum, whereas the lowest sensitivity, of 937.8 nm/RIU, was achieved for a cladding-removed multimode optical fiber (CRMOF) with a 2% aluminum-doped AZO film. In a similar study, an AZO-coated fiber optic LMR sensor was demonstrated experimentally to further enhance the sensitivity of the LMR sensor, and the results demonstrated a sensitivity value of 2280 nm/RIU in the near-IR wavelength region [67]. A detailed summary of a ZnO-coated fiber optic LMR sensor, with its configuration and important applications, is presented in Table 2. 

### 4.4. Titanium Oxide (TiO_2_)

It was noted that when an additional layer of TiO_2_ thin film was deposited over an AZO-coated tapered-tip fiber, it significantly enhanced the performance of LMR-based refractometers [72]. However, TiO_2_ is itself a unique CMO for the evolvement of fiber optic LMR sensors because of its higher value of RI and lower value of extinction coefficient (Figure 13), which satisfy the resonance condition to generate LMRs. Moreover, the thin-film deposition of TiO_2_ is easily achievable by employing several thin-film deposition techniques such as LbL and sputtering, which further enhance the capability of TiO_2_ in RI and biosensing applications. 

The first theoretical and experimental investigation of a TiO_2_ thin film to be used as a LMR active material to generate multiple resonances was reported in 2010 [40]. The study consisted of bilayers of TiO_2_/PSS nanocoating over a multimoded optical fiber, using LbL deposition technique that provided a smooth and homogenous coating, which also leads to the generation of several resonances in a wide wavelength range, from ultraviolet to the IR region of the optical spectrum. At the output end of the optical fiber, the transmission spectra were measured as a function of the wavelength for the different thicknesses of the TiO_2_/PSS bilayers. During this procedure, the effective bilayer thickness was measured by dividing the coating thickness by the number of bilayers coated after the end of the deposition process, and the maximum value of the RI sensitivity turns out to be 2872.73 nm/RIU, with a 460 nm coating thickness. In a similar kind of study, Zamarreno et al. presented the fabrication techniques for the TiO_2_/PSS bilayers when coated over a side-polished D-shaped fiber [145]. It was reported that the number of produced LMRs are thickness-dependent and very sensitive, so a small change in the SMRI permits their employment as refractometers. Additionally, the results were obtained for both TM- and TE-polarized light, which results in a LMR shift of 226 nm and 56 nm for the first- and second-order LMRs, respectively. Later on, Villar et al. discussed the optimization of various parameters to develop highly sensitive fiber optic LMR sensors utilizing various nanocoatings [119]. A thin film of TiO_2_ was also used as an additional layer, over ITO and AZO, to theoretically enhance the performance of the designed LMR sensors [72,120]. It was revealed that an additional layer of TiO_2_ over ITO results in a two-fold increment in sensitivity, when compared with a LMR sensor without TiO_2_, and this value can be tuned just by varying the total bilayer thickness and the bilayer ratio of ITO/TiO_2_. Further, TiO_2_-based LMR sensors were utilized in RH sensing by considering thin films of TiO_2_ and PSS, which were coated over a single-mode–multimode–single-mode optical fiber device and a LMR-based device, simultaneously [121]. The study performed the utilization of designed fiber configuration in RI and RH sensing. From results it was attained that the RI and RH sensitivities corresponding to single-mode–multimode-single-mode configuration turn out to be 142 nm/RIU and 0.3 nm/RH%; whereas the corresponding values for a LMR-based device were observed to be 955 nm/RIU and 3.54 nm/RH%, respectively. After this, a highly sensitive ammonia (NH_3_) sensor was reported with a nanocoating of TiO_2_ film and porphyrin as a functional material over a tapered-core fiber optic probe [73]. It was revealed that a change in the RI of the coated material, due to the interaction of porphyrin with NH_3_, resulted in a variation in the LMR wavelength. Thus, by measuring the LMR shift in the transmission spectrum, a low concentration of NH_3_ in water (0.1 ppm) can be measured with a response time of 30 s and a detection limit of 0.16 ppm. The sensitivity of LMR-based optical fiber refractometers using TiO_2_ was further enhanced to 4122 nm/RIU, for the SMRI ranging from 1.333 to 1.398 [122]. The sensor utilized a D-shaped optical fiber with the deposition of a TiO_2_ film over its polished surface, with a residual thickness of 72 µm, and then it was placed in contact with the sensing medium of a varied RI. More recently, a novel LMR-based pressure sensor was realized theoretically on a microstructure optical fiber [123]. The designed fiber configuration uses the deposition of a TiO_2_/HfO_2_/rubber polymer over an exposed core PCF, and the air holes were assumed to be filled with liquid. The maximum RI sensitivity value of the LMR sensor was 67,000 nm/RIU, in the RI ranging from 1.33 to 1.39, and, more importantly, the pressure sensitivity for the bilayers’ TiO_2_/HfO_2_ configuration comes out to be 5 µm/MPa. A thorough summary of the TiO_2_-nanocoating-based fiber optic LMR sensor and its few applications are presented in Table 2.

### 4.5. Tin Oxide (SnO_2_)

Tin oxide (SnO_2_), also known as stannic oxide, has been recently studied for designing LMR-based FOSs because of its ability to generate LMRs. In the literature, it was revealed that the nanocoating of SnO_2_ is very sensitive to any change in the humidity condition surrounding it; therefore, SnO_2_ is utilized as a coating material for humidity sensors [55]. A fabricated optical fiber humidity sensor was verified in a climate chamber to detect the response towards a percentage change in RH. It was observed that when the concentration of RH varies from 20–90%, a variation of more than 130 nm was noted in the RW, which gives rise to a maximum sensitivity value of 1.9 nm/%RH. Apart from RH sensing, SnO_2_-based LMR sensors were also utilized for NH_3_ gas sensing because the interaction of SnO_2_ with NH_3_ results in the variation of its RI [122,123,146]. Further, a fiber optic LMR sensor, with the utilization of a SnO_2_ thin film in RI-sensing applications, provides a high sensitivity value of 5390 nm/RIU, which is much higher in comparison with the values of other LMR sensors [42]. The reason for this higher value is because of the higher RI value of SnO_2_ when compared with other CMOs such as In_2_O_3_ and TiO_2_. In a recent investigation of a SnO_2_-based LMR refractometer, by means of a D-shaped optical fiber, it exhibited a sensitivity of more than 106 nm/RIU, as measured by a picometer resolution detector [124]. The study also considered the effect of coated film thickness and the effect of using both TM- and TE-polarized light on the sensitivity of the designed LMR sensor. Moreover, a fiber optic LMR sensor was fabricated for the detection of arsenic, by utilizing SnO_2_ thin film/NPs coated over a cladding-etched optical fiber core [125]. The experimental results demonstrated that the highest sensitivity value, of 1.31 nm/μgL^−1^, can be achieved near a zero concentration of arsenic (III), which then starts decreasing with an increase in concentration because of the presence of a finite number of active sites available on the surface. Further, the other performance parameters of the designed sensor were also measured such as the stability, response time of the sensor, and selectivity of the fiber optic probes. More recently, Vicente et al. proposed a SnO_2_-coated LMR sensor, by utilizing a novel optical fiber structure, and attained the RI sensitivity values of 7364.93 nm/RIU and 708.57 nm/RIU, corresponding to the first and second LMRs, respectively [126]. The designed sensor was further utilized for the detection of IgG concentration, with a concentration range in between 1–40 mg/L with a detection limit of 0.6 mg/L. A full summary of the SnO_2_-coated fiber optic LMR sensor with its important sensing applications is presented in Table 2.

### 4.6. Polymers

In the literature, it was found that apart from CMOs there are certain polymers that satisfy the generation condition of LMR such as polyallylamine hydrochloride (PAH), polyacrylic acid (PAA), and polystyrene sulfonate (PSS). It was reported that the LbL deposition of PAH/PAA results in the generation of LMRs, whereas PSS was utilized in a LbL combination with TiO_2_, in order to attain resonances [145,147]. The ionic conductivity of these polymers can be varied depending upon several parameters such as the RI of the sensing medium, the pH, and the humidity; due to these parameters, polymers have found their usefulness in various RI, humidity, and pH sensors [128,148,149]. As a result, when a light propagating through a waveguide interacts with a polymeric film, its conductivity changes, hence, resulting in the excitation of the various lossy mode inside it. Here, we will discuss some fiber optic LMR sensors that have utilized the polymer either as a LMR active material or a recognition element. A fiber optic LMR sensor utilizing a thin polymeric film for pH sensing was first reported by Zamarreno et al. [127]. Their designed scheme used a thin film of PAH and PAA as a coating, with a thickness that varied in accordance with the pH of the solution. For sensor fabrication, a cladding-etched portion of fiber was attached to a U-shaped holder, to avoid any bending or damage to the sensing region. After this, a thin film of polymer was coated over the fiber core using the LbL electrostatic self-assembly method. For the designed sensor, the absorption spectra were measured as a function of the wavelength, when the number of PAH/PAA bilayers were increased from 1 to 100. The study concluded that the designed LMR-based pH sensor possessed a maximum sensitivity of 0.027 pH units/nm, for pHs of 3 and 6, with an accuracy of ±0.001 pH units. Soon after this, polymeric films were utilized in the fabrication of fiber optic LMR sensors for the detection of immunoglobulin G [147]. The study involved the deposition of a PAH/PSS polymeric film on a cladding-etched optical fiber core using the LbL deposition technique, and the results demonstrated that a shift of 10 nm in the LMR wavelength is detected, while sensing the immunoglobulins with a response time of 12 min and a concentration of 50 µg/mL. In another work, the same authors designed a high-sensitivity fiber optic pH sensor based on the LMR technique, by utilizing a PAH and PAA thin polymeric film coated over a tapered fiber core [70]. For the designed pH sensor, the maximum wavelength shift reached up to 200 nm for a pH ranging from 4.0 to 6.0 with a resolution of 0.05 units of pH, and the corresponding response time reached 60 s. Another pH sensor was reported by Rivero et al., utilizing AuNPs into a polymeric LbL coating [49]. They reported that the designed fiber optic LMR sensor showed a dynamic range of 134.7 nm for a pH between 4.0 and 6.0, whereas the LSPR band showed a dynamic range of 1.50 nm only within the same pH range. Moreover, pH sensitivity corresponding to LMR sensors was observed as 67.35 nm/pH unit, which is much higher in comparison with the sensitivity obtained from LSPR (0.75 nm/pH unit). Another sensing application of thin polymeric-film-based LMR sensors is the RH, which is also studied in the literature. Rivero et al. fabricated an optical fiber RH sensor utilizing a thin polymeric film doped with AgNPs based on the LMR and LSPR techniques [128]. It was revealed that a polymeric coating with a combination of AgNPs is more sensitive to RH changes, when compared to the results with a polymeric layer only. Moreover, the monitoring of human breathing showed a high sensitivity, of 0.0943 nm/RH%, for the designed fiber optic LMR sensor. In a similar study, the sensitivity values corresponding to the first and second LMRs were 0.56 nm/RH% and 0.3 nm/RH%, respectively, for RH sensing [147]. In 2017, Urrutia et al. presented a polymeric-film-based LMR sensor, in which Au nanorods were embedded in a thin polymeric film by employing the LbL self-assembly deposition technique [130]. The combined layers of polymeric film and Au nanorods, with different thicknesses, were coated over a cladding-etched portion of the optical fiber, and the spectral response was monitored. The fabricated LMR sensor was tested in a chamber having a RH range between 20–90%, and it was observed that a high sensitivity of 11.2 nm/%RH can be achieved with LMR sensing. A thorough summary of thin polymeric-film-coated fiber optic LMR sensors with their significant sensing applications is presented in Table 2.

### 4.7. Other Materials

Until now, we have presented a detailed account of the most commonly employed LMR supporting materials for the fabrication of FOSs and their utilization in various sensing schemes. This has included several CMOs and polymeric films that possess several important characteristics, in terms of their dispersion relation, and thin-film deposition is also easily achievable for these materials. Apart from these materials, there are some significant oxide materials that have gained considerable attention in the field of fiber optic LMR sensing such as indium-gallium–zinc–oxide (IGZO), graphene oxide (GO), copper oxide (CuO), hafnium dioxide (HfO_2_), zirconium dioxide (ZrO_2_), aluminum oxide (Al_2_O_3_), niobium pentoxide (Nb_2_O_5_), tantalum oxide (Ta_2_O_5_), etc. Tien et al. reported a LMR-based liquid salinity sensor utilizing a thin film of IGZO over a D-shaped single-mode optical fiber [48]. An IGZO thin film with a thickness of 100 nm was coated over the side-polished surface of a fiber via the RF magnetron sputtering technique. To measure the RW shift in the transmission spectra, different solutions of liquid salinities were prepared and placed around the sensing layer. For the designed liquid salinity sensor, the highest sensitivity was 0.80 nm/SU (salinity unit). In a similar study for IGZO, a maximum sensitivity of 12,929 nm/RIU was reported for the SMRI ranging from 1.39 to 1.42 [80]. It is clearly visible that with an increase in the thickness of the IGZO film, the RW shifted to higher wavelength values, resulting in better sensitivity values. Another important material that has been utilized recently in fiber optic LMR sensing is GO. A LMR-based FOS employing a thin GO film was reported, in 2019, by Hernaez et al. [131]. In that study, thin films of GO and polyethyleneimine (PEI) were deposited over a fiber core via the LbL deposition technique. In addition, the study involved the fabrication and characterization of two different devices employing 8 and 20 bilayers of thin films. The corresponding values of sensitivity with eight bilayers were 12,247 nm/RIU and 12,460 nm/RIU, for the increased and decreased SMRIs, respectively, with a resolution of 8 × 10^−5^ RIU. Similarly, the equivalent values of sensitivity for 20 bilayers were 2517 nm/RIU and 2631 nm/RIU, respectively, with a resolution of 3.8 × 10^−4^ RIU. In another study, a copper(II) oxide based fiber optic LMR sensor was proposed for RI-sensing applications [132]. The study revealed that a CuO thin film is capable of exciting the LMR in both the visible and near-IR wavelength regions, which also gives rise to better sensing capabilities. After optimizing, the certain parameters’ highest sensitivity for the designed LMR sensor was 7324 nm/RIU, which means that a CuO-fabricated LMR sensor can also be found to be useful in sensing applications. Additionally, HfO_2_- and ZrO_2_-based fiber optic LMR sensors were fabricated over a fiber core via the ALD technique and showed satisfactory sensing properties [133]. 

## 5. Conclusions and Future Prospects

This review article presented a comprehensive (but not critical) review of fiber optic LMR sensors. We started with a brief introduction of the evolution of optical sensing, origin of lossy guidance in a waveguide through a thin conducting film, and developmental history of fiber optic LMR sensors. We discussed the characteristics of LMRs and the setup of and configuration for designing LMR sensors and also differentiated LMR from the well-established SPR phenomenon. We also provided a detailed explanation of the geometry/structure of the optical fiber probes supporting LMR. Finally, we made a detailed analysis of LMR-supported coated materials and their utilization in various physical, chemical, and biosensing applications. 

Since LMR-based FOSs are still an emerging field of research, it should be noted that a lot of studies have been performed during the last decade. This shows that LMR sensing has stood up as a tough competitor to well-established optical sensing technologies such as SPR, LSPR, and long-period fiber gratings. As discussed in previous sections, different configurations and geometries of optical fibers have been realized, which revealed that an adequate choice of fiber structure permits us to attain the reduced width of the LMR spectrum and, hence, an increased resolution of the designed sensor. However, optical fiber structures such as tapered tip, photonic crystal fiber, hollow core fiber, and bend fibers are still unexplored regarding experimental LMR sensing and, hence, need to be explored more in the future because these configurations have already been studied for SPR sensing. Out of several coated materials, ITO-based fiber optic LMR sensors produce a better performance and also become interesting because of their dual nature of generating resonances. However, there is a wide scope in the future to further explore the materials that are suitable for generating LMRs and their deposition techniques over a fiber substrate, to characterize the best-suited LMR sensor. At present, fiber optic LMR sensors are mainly utilized in refractometer sensing because finding a decent refractometer creates the need to design chemical sensors and biosensors. Apart from this, fiber optic LMR sensors have also been exploited in gas, RH, and voltage sensing applications. However, LMR-based FOSs can be better applied in biotechnology, food safety measurements, medical diagnostics, temperature measurements, and environmental monitoring. We believe that the present review article is worthy enough for scholars/scientists and can be useful for them in better understanding fiber optic LMR sensors and their development status. 

## Figures and Tables

**Figure 1 micromachines-13-01921-f001:**
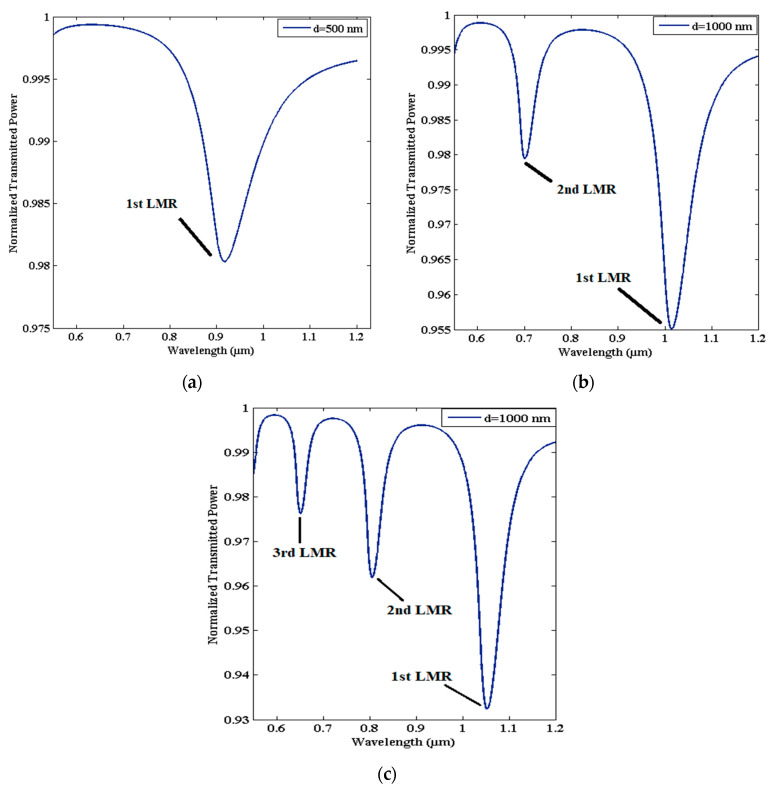
Transmission spectra of LMR-based FOS coated with AZO, having thicknesses of (**a**) 500 nm, (**b**) 1000 nm, and (**c**) 1500 nm. Other parameters that were used: SMRI = 1.333, length of sensing region = 0.5 cm, numerical aperture = 0.22, and core diameter = 600 µm.

**Figure 2 micromachines-13-01921-f002:**
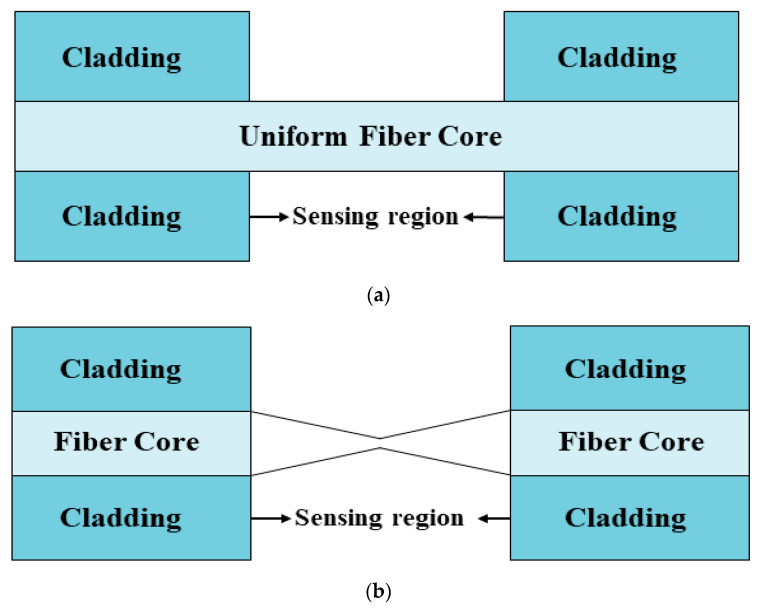
Schematic view of LMR-based (**a**) uniform-core fiber optic probe, (**b**) tapered-core fiber optic probe, and (**c**) uniform-core tapered fiber optic probe.

**Figure 3 micromachines-13-01921-f003:**
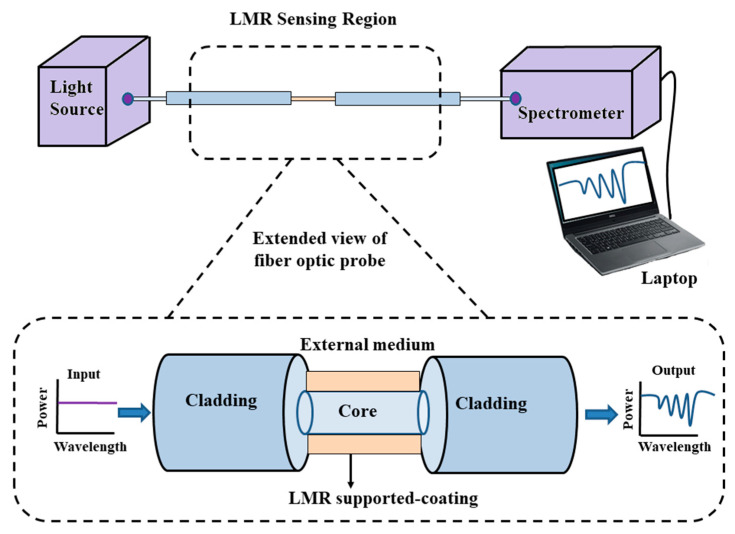
A detailed view of transmission setup of fiber optic LMR sensor.

**Figure 4 micromachines-13-01921-f004:**
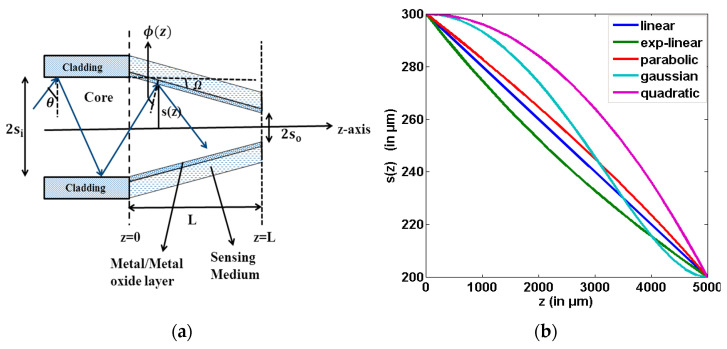
(**a**) Structural view of used tapered-core fiber optic probe; (**b**) variation of the taper radius *s*(*z*) with distance *z* for five different taper profiles. Graphical presentation of sensitivity with TR along with five taper profiles for (**c**) first and (**d**) second LMR of ITO, respectively. Reproduced with permission from [74], copyright IOP Publishing.

**Figure 5 micromachines-13-01921-f005:**
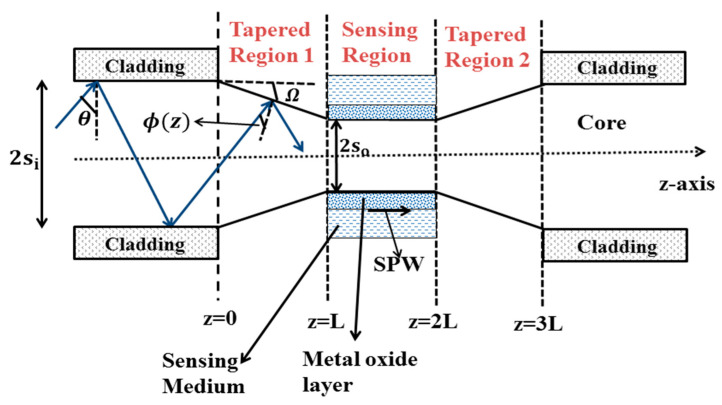
Schematic diagram of proposed LMR-based fiber probe having uniform fiber core placed between two identical tapered regions. Reproduced with permission from [75], copyright IOP Publishing.

**Figure 6 micromachines-13-01921-f006:**
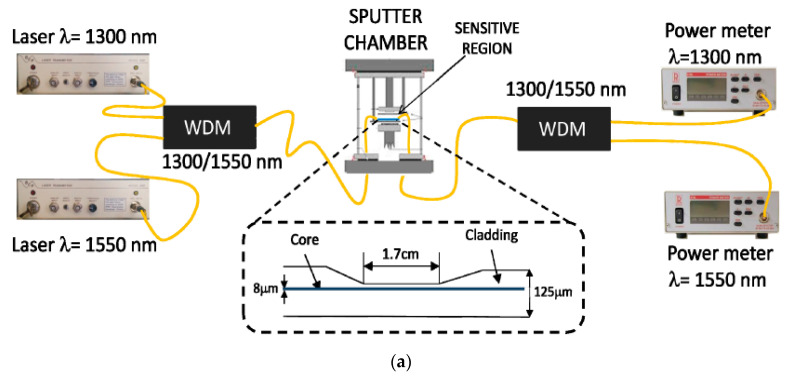
(**a**) Schematic view of experimental setup for the fabrication process of ITO and D−shaped fiber; (**b**) variation of output optical power with deposition time (coating thickness) at 1300 nm and 1550 nm. Reprinted with permission from [77], copyright The Optical Society.

**Figure 7 micromachines-13-01921-f007:**
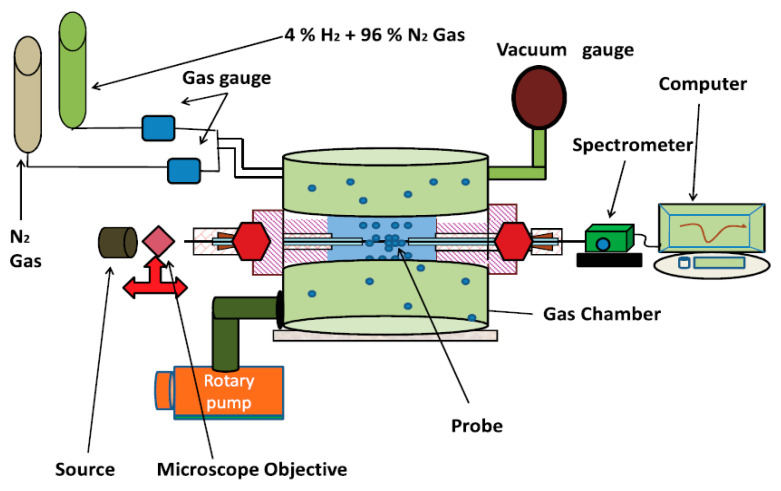
A schematic view of experimental gas-sensing system for designing LMR-based hydrogen gas sensor. Reproduced with permission from [57], copyright IOP Publishing.

**Figure 8 micromachines-13-01921-f008:**
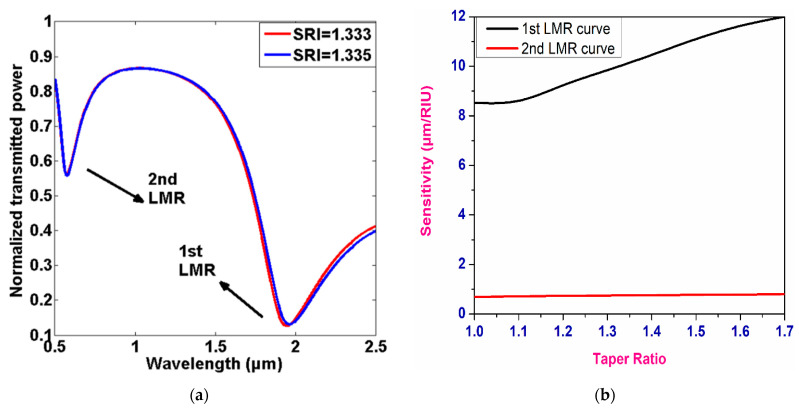
(**a**) Transmission curves for LMR-based FOS coated with ITO film at taper ratio 1.0, for two different SMRIs. (**b**) Sensitivity evaluation for first and second LMRs with taper ratio. Reproduced with permission from [74], copyright IOP Publishing.

**Figure 9 micromachines-13-01921-f009:**
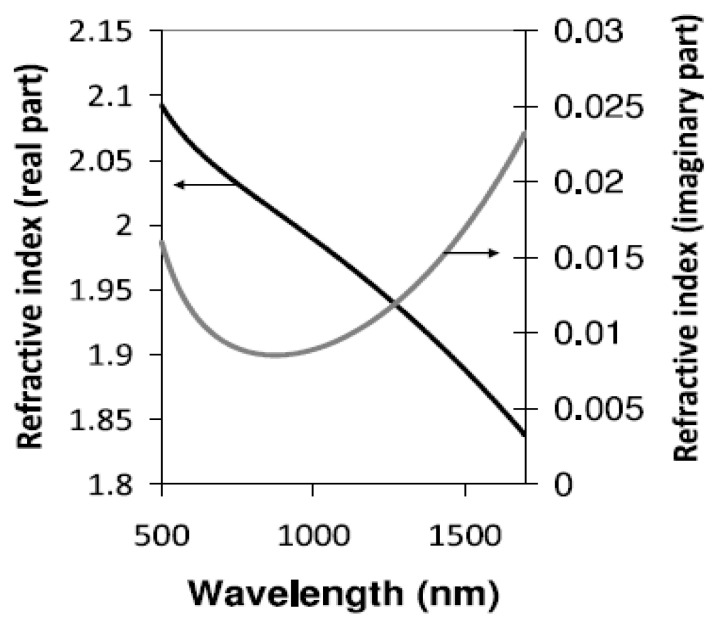
The variation of real and imaginary parts of RI of In_2_O_3_, with wavelength. Reproduced with permission from [136], copyright IOP Publishing.

**Figure 10 micromachines-13-01921-f010:**
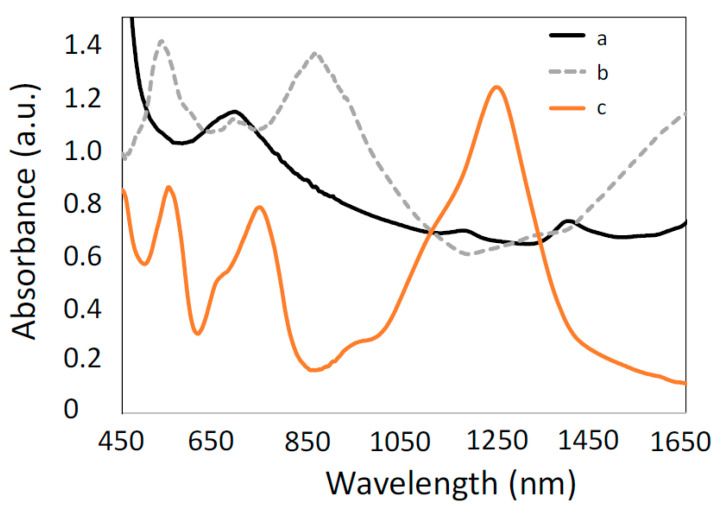
Absorption peaks observed from the device having different coating thickness ((a) 70 nm, (b) 300 nm, and (c) 690 nm). Reprinted with permission from [114].

**Figure 11 micromachines-13-01921-f011:**
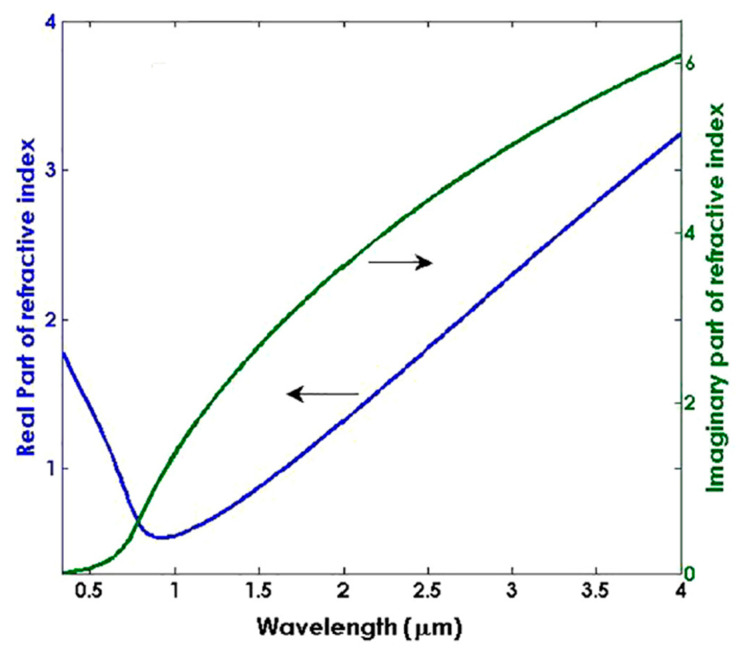
Dispersion curves for ZnO. Reprinted with permission from [84], copyright The Optical Society.

**Figure 12 micromachines-13-01921-f012:**
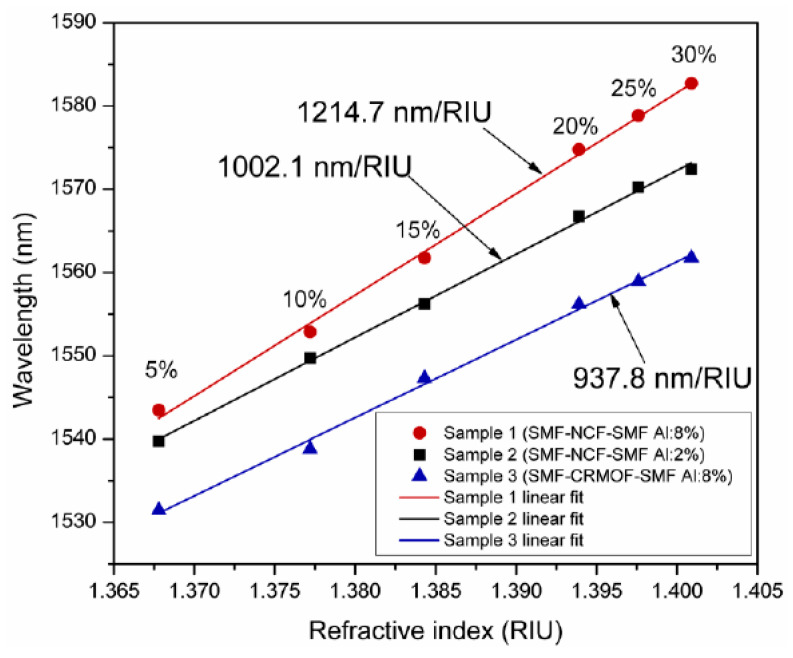
Variation of resonance wavelength with different concentrations of isopropyl alcohol/glycerine, for the proposed fiber structure. Reprinted with permission from [118].

**Figure 13 micromachines-13-01921-f013:**
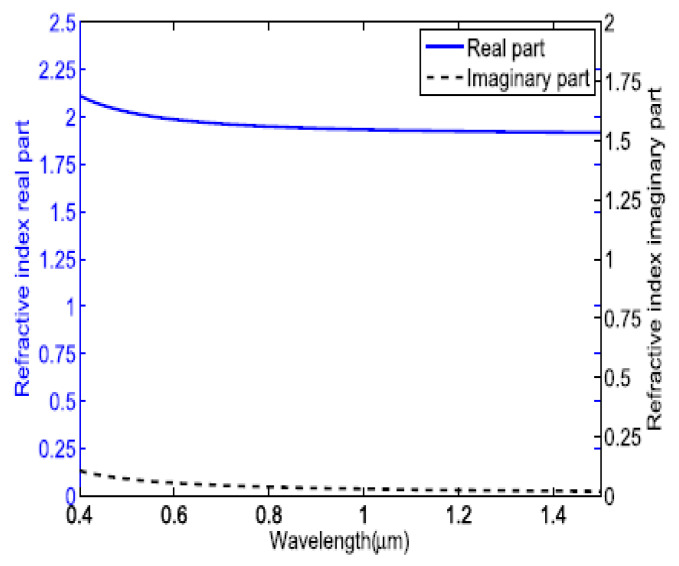
Dispersion curves for TiO_2_/PSS. Reprinted with permission from [119], copyright The Optical Society.

**Table 1 micromachines-13-01921-t001:** Resonance condition for thin-film-coated optical waveguides, in which $\varepsilon_{m}^{\prime }$ and $\;\varepsilon_{m}^{\prime \prime }$ denote the real and imaginary parts of coated material, respectively.

Type of Resonance	Resonance Conditions $\left(\mathrm{If}\;\varepsilon_{m}=\varepsilon_{m}^{\prime }+i\;\varepsilon_{m}^{\prime \prime }\right)$
Surface plasmon resonance (SPR)	$\varepsilon_{m}^{\prime }<0$ $\left|\varepsilon_{m}^{’}|>\;|\varepsilon_{m}^{\prime \prime }\right|$ $\left|\varepsilon_{m}^{\prime }|>|\varepsilon_{s}\right|$
Lossy mode resonance (LMR)	$\varepsilon_{m}^{\prime }>0$ $\left|\varepsilon_{m}^{\prime }|>\;|\varepsilon_{m}^{\prime \prime }\right|$ $\left|\varepsilon_{m}^{’}|>|\varepsilon_{s}\right|$

**Table 2 micromachines-13-01921-t002:** Detailed summary of fiber optic LMR sensors for a wide range of coated materials, several geometries, and their utilization in various sensing schemes.

Fiber Configuration	Type of Study	Coated Material	Sensing Parameter	Range of RI	Maximum Sensitivity	Spectral Range (nm)	Ref.
200 µm CRMOF	Experimental	ITO	RI	1.321–1.436	3125 nm/RIU	900–1500	[38]
200 µm CRMOF	Experimental	ITO and PAA–PAH	RH	1.32–1.40	0.833 nm/%RH	500–1700	[99]
200 µm CRMOF	Experimental	ITO and PVDF and ITO	Voltage	-	0.4 nm/V	500–1000	[100]
400 µm CRMOF	Experimental	ITO	Voltage	1–1.4296	10 mVs^−1^	400–900	[101]
600 µm CRMOF	Experimental	ITO and ITO NPs	H_2_ gas	-	0.71 nm/ppm	350–700	[57]
200 µm CRMOF	Experimental	IT/SnO_2_	Turbine oil	-	0.27 × 10^−3^ nm/h	1000–1700	[110]
400 µm CRMOF	Experimental	ITO	Ketoprofen	-	1400 nm/M	400–1000	[111]
Tapered	Theoretical	ITO/AZO	RI	1.33–1.34	12,005 nm/RIU	500–2500	[74]
200 µm CRMOF	Experimental	In_2_O_3_	RI	1.333–1.392	4068 nm/RIU	500–1700	[113]
200 µm CRMOF	Experimental	In_2_O_3_	RI	1.32–1.37	4926 nm/RIU	400–1700	[39]
200 µm CRMOF	Experimental	In_2_O_3_ and PAH–PSS	RH	1.32–1.40	0.935 nm/%RH	500–1700	[99]
200 µm CRMOF	Experimental	In_2_O_3_	RI	1.332–1.407	5680 nm/RIU	400–1700	[114]
600 µm CRMOF	Experimental	In_2_O_3_	RI	1.33–1.39	2937 nm/RIU	400-700	[115]
600 µm CRMOF	Experimental	ZnO and ZnO NPs	H_2_S gas	-	1.49 nm/ppm	400–1000	[43]
600 µm CRMOF	Experimental	ZnO and ZnO nanorods	H_2_S gas	-	4.14 nm/ppm	350–650	[116]
600 µm CRMOF	Experimental	ZnO and ZnO–PPY	Cortisol	-	12.86 nm/log	350–900	[51]
U-shaped	Theoretical and Experiment	ZnO	RI	1.33–1.42	900 nm/RIU	350–500	[76]
600 µm CRMOF	Experimental	ZnO and MoS_2_	Urinary p-cresol	-	11.86 nm/µM	300–700	[83]
CRMOF	Theoretical	ZnO and HfO_2_	Pressure	1.33–1.45	2 µM/MPa	400–1000	[117]
SM-MM-SM	Experimental	AZO	RI	1.365–1.40	1214.7 nm/RIU	400–2500	[118]
200 µm CRMOF	Experimental	AZO	RI	1.33–1.45	2280 nm/RIU	450–900	[67]
200 µm CRMOF	Theoretical and Experiment	TiO_2_ and PSS	RI	1.33–1.43	2872.73 nm/RIU	400–1500	[40]
200 µm CRMOF	Theoretical and Experiment	TiO_2_ and PSS	RI	1.321–1.421	4000 nm/RIU	400–1500	[119]
CRMOF	Theoretical	ITO and TiO_2_	RI	1.331–1.436	23,000 nm/RIU	400–2000	[120]
Tapered	Theoretical	AZO and TiO_2_	RI	1.34–1.45	9000 nm/RIU	400–800	[72]
200 µm CRMOF	Experimental	TiO_2_ and PSS	RH	-	3.54 nm/%RH	1150–1650	[121]
Tapered	Experimental	TiO_2_ and Porphyrin	NH_3_	-	10,000 nm/ppm	600–1000	[73]
D-shaped	Experimental	TiO_2_	RI	1.333–1.398	4122 nm/RIU	900–1700	[122]
D-shaped	Theoretical	TiO_2_ and HfO_2_ and Rubber Polymer	RI	1.33–1.39	67,000 nm/RIU	600–2000	[123]
Etched SMF	Theory + Experiment	SnO_2_	RH	-	1.9 nm/%RH	1300–1700	[55]
200 µm CRMOF	Experimental	SnO_2_	RI	1.33–1.41	5390 nm/RIU	450–1650	[42]
D-shaped	Experimental	SnO_2_	RI	1.441–1.449	106 nm/RIU	1150–1650	[124]
CRMOF	Experimental	SnO_2_ NPs and α-Fe@Sn CS	Arsenite	-	1.31 nm/µgL^−1^	350–650	[125]
MM-coreless-MM	Experimental	SnO_2_	IgG	-	0.6 mg/L	400–1600	[126]
200 µm CRMOF	Theory + Experiment	PAH–PAA	pH	-	36.67 nm/pH, pH = 3 to 6	400–1000	[127]
200 µm CRMOF	Experimental	PAH–PAA and AuNPs	pH	-	67.35 nm/pH, pH = 4 to 6	450–1000	[49]
200 µm CRMOF	Experimental	PAH–PAA and AgNPs	RH	-	0.0943 nm/%RH	400–1100	[128]
200 µm CRMOF	Experimental	PAH–PAA	RH	-	0.56 nm/%RH	500–800	[129]
200 µm CRMOF	Experimental	PAH–GNR@PSS	RH	-	11.2 nm/%RH	400–1000	[130]
D-shaped	Experimental	IGZO	Salinity	-	0.8 nm/SU	900–1700	[56]
D-shaped	Experimental	IGZO	RI	1.39–1.42	12929 nm/RIU	1150–1650	[80]
200 µm CRMOF	Experimental	GO–PEI	RI	1.33–1.42	12460 nm/RIU	400–900	[131]
200 µm CRMOF	Experimental	CuO	RI	1.3335–1.4075	7324 nm/RIU	400–1550	[132]
400 µm CRMOF	Experimental	ZrO_2_	RI	1.41–1.43	880 nm/RIU	400–900	[133]

## Data Availability

Not applicable.

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
