# Peer review of "Recent Advances in Lossy Mode Resonance-Based Fiber Optic Sensors: A Review"

_micromachines, 2022, doi:10.3390/mi13111921_

Round 1
Reviewer 1 Report
This is a good review and can be published in micromachines. One minor suggestion is to add some comparison with other optical sensors eg. SPR and LSPR sensors. I have shared below some references to look for:
1. Tu, M. H., T. Sun, and K. T. V. Grattan. "LSPR optical fibre sensors based on hollow gold nanostructures." Sensors and Actuators B: Chemical 191 (2014): 37-44.
2.Chen, Yuzhi, et al. "Fiber-optic urine specific gravity sensor based on surface plasmon resonance." Sensors and Actuators B: Chemical 226 (2016): 412-418.
3. Bhalla, Nikhil, et al. "Nanoplasmonic biosensor for rapid detection of multiple viral variants in human serum." Sensors and Actuators B: Chemical 365 (2022): 131906.
4. Gomaa, Mahmoud, Abeer Salah, and Gamal Abdel Fattah. "Superior enhancement of SPR fiber optic sensor using laser sensitized dip-coated graphene gold nanocomposite probes." Optics & Laser Technology 157 (2023): 108644.
Author Response
Thank you very much for your appreciation and comment. We have revised our manuscript accordingly and the response to your query is as follows:
Comment: This is a good review and can be published in micromachines. One minor suggestion is to add some comparison with other optical sensors e.g., SPR and LSPR sensors.
Response: Thank you for this comment. We have added a short paragraph about LSPR sensors in the revised manuscript (Section 2.1).

Reviewer 2 Report
The present review article covers a comprehensive analysis of LMR based sensor covering the fundamentals, geometrical configurations, and sensing applications. The article is written well and covered a wide range of references for LMR based sensors. The manuscript has the potential to be published in the current form.
However, I have a few minor comments before the publication:
1. There are several review articles for LMR based sensors are already present in the literature. Authors should provide the necessity of this review article in the introduction section.
2. Section 2.2: The comparison between SPR and LMR is too vague. A more detailed comparison should be made in terms of sensitivity, FOM and as well as the superiority of SPR and LMR based on the application.
Author Response
Thank you very much for thoroughly reviewing our manuscript. We have revised our manuscript accordingly and point-by-point response to your queries are as under:
- Query: There are several review articles for LMR based sensors are already present in the literature. Authors should provide the necessity of this review article in the introduction section.
Response: The introduction section in the revised manuscript has been modified.
- Query: Section 2.2: The comparison between SPR and LMR is too vague. A more detailed comparison should be made in terms of sensitivity, FOM and as well as the superiority of SPR and LMR based on the application.
Response: Section 2.2 in the revised manuscript has been modified.
